# Effect of Co-Inoculation of *Bacillus* sp. Strain with Bacterial Endophytes on Plant Growth and Colonization in Tomato Plant (*Solanum lycopersicum*)

**Ahsanul Salehin** [1], **Ramesh Raj Puri** [2], **Md Hafizur Rahman Hafiz** [3,4] **and Kazuhito Itoh** [1,3,*]

1    The United Graduate School of Agricultural Sciences, Tottori University, 4-101 Koyama-Minami, Tottori 680-8553, Japan; ujanrijvi224@gmail.com
2    Agriculture Victoria Research, Department of Jobs, Precincts and Regions, 110 Natimuk Road, Horsham, VIC 3400, Australia; ramesh.rajpuri@ecodev.vic.gov.au
3    Faculty of Life and Environmental Sciences, Shimane University, 1060 Nishikawatsu, Matsue 690-8504, Japan; hafizhstu@hotmail.com
4    Department of Crop Physiology and Ecology, Hajee Mohammad Danesh Science and Technology University, Dinajpur 5200, Bangladesh
*    Correspondence: itohkz@life.shimane-u.ac.jp; Tel.: +81-852-32-6521

**Abstract:** Colonization of a biofertilizer *Bacillus* sp. OYK strain, which was isolated from a soil, was compared with three rhizospheric and endophytic *Bacillus* sp. strains to evaluate the colonization potential of the *Bacillus* sp. strains with a different origin. Surface-sterilized seeds of tomato (*Solanum lycopersicum* L. cv. Chika) were sown in the sterilized vermiculite, and four *Bacillus* sp. strains were each inoculated onto the seed zone. After cultivation in a phytotron, plant growth parameters and populations of the inoculants in the root, shoot, and rhizosphere were determined. In addition, effects of co-inoculation and time interval inoculation of *Bacillus* sp. F-33 with the other endophytes were examined. All *Bacillus* sp. strains promoted plant growth except for *Bacillus* sp. RF-37, and populations of the rhizospheric and endophytic *Bacillus* sp. strains were 1.4–2.8 orders higher in the tomato plant than that of *Bacillus* sp. OYK. The plant growth promotion by *Bacillus* sp. F-33 was reduced by co-inoculation with the other endophytic strains: *Klebsiella* sp. Sal 1, *Enterobacter* sp. Sal 3, and *Herbaspirillum* sp. Sal 6., though the population of *Bacillus* sp. F-33 maintained or slightly decreased. When *Klebsiella* sp. Sal 1 was inoculated after *Bacillus* sp. F-33, the plant growth-promoting effects by *Bacillus* sp. F-33 were reduced without a reduction of its population, while when *Bacillus* sp. F-33 was inoculated after *Klebsiella* sp. Sal 1, the effects were increased in spite of the reduction of its population. *Klebsiella* sp. Sal 1 colonized dominantly under both conditions. The higher population of rhizospheric and endophytic *Bacillus* sp. in the plant suggests the importance of the origin of the strains for their colonization. The plant growth promotion and colonization potentials were independently affected by the co-existing microorganisms.

**Keywords:** OYK; PGPR; *Bacillus* sp.; tomato (*Solanum lycopersicum*); endophytes; co-inoculation; colonization





## 1. Introduction

Plant growth-promoting rhizobacteria (PGPR) are becoming more widely accepted in intensive agriculture to enhance sustainable agricultural production in various parts of the world [1]. PGPR contain a diverse range of bacteria and several mechanisms have been proposed though they are not fully understood [2]. In sustainable agricultural practices using PGPR, inoculation techniques for their colonization at the rhizosphere is critical [3]; therefore, a further understanding of the interactions of PGPR with plant and indigenous rhizobacteria is essential.

*Bacillus* spp. have been recognized as one of the most important PGPR and widely used for sustainable agriculture as biofertilizers and/or antagonists against plant diseases [4–8].

*Bacillus* spp. have also received considerable attention because of their benefits over other PGPR in producing stable formulations [6,9] and stability in rhizosphere soil in semi-arid deserts [10]. In addition, *Bacillus* spp. exhibit a significant reduction in disease incidence on various crops by inducing systemic resistance [11,12] and by forming biofilm on root surfaces [13].

In our previous study, when the commercial biofertilizer OYK consisting of the *Bacillus* sp. strain was applied to sweet potato, no significant plant growth-promoting effect was observed, and the inoculated *Bacillus* sp. strain was not detected in the plant tubers. The possible reasons were due to competition of the inoculant against indigenous rhizobacteria and endophytes, and a lack of endophytic potential of the inoculant, which was originally isolated from soil [14]. As many endophytic *Bacillus* strains have been reported in several plants [15–22], it is assumed that endophytic bacteria have some colonization strategies in interaction with plants.

In addition to the individual colonizing ability of PGPR, interactions with other co-existing bacteria would be important to determine the colonization and plant growth-promoting potential. Synergetic effects of the inoculation with the other PGPR have been reported in maize [23], cotton [24], ryegrass [25], strawberry [26], and cucumber [27]. On the other hand, negative interactions with co-existing bacteria should also be considered. They inhibited the colonization of inoculants in sugarcane [28] and reduced the plant growth-promoting effects in tomato plant [29,30].

For efficient and practical use of PGPR, it is essential to understand its colonizing behavior and abilities to compete with co-existing bacteria. Though several studies have been reported on the effects of co-inoculation with multiple bacteria on plant growth, their effects on colonization have not been extensively studied yet. The aim of this study was to evaluate the colonization properties of *Bacillus* sp. OYK, which was isolated from a soil, in relation to its origin by comparing it with those of the other *Bacillus* sp. strains isolated from plant endosphere and rhizosphere, and then to elucidate the effects of co-inoculation of the endophytic *Bacillus* sp. strain with the other endophytes on their colonization and plant growth-promoting activities.

## 2. Materials and Methods

### 2.1. Bacterial Strains

In addition to *Bacillus* sp. OYK, three strains of *Bacillus* sp.: two strains (*Bacillus* sp. RF-12 and RF-37) isolated from the rhizosphere of sweet potato and another one (*Bacillus* sp. F-33) as an endophyte of the same plant cultivated in Japan [16], and three strains of endophytes: *Herbaspirillum* sp. Sal 6, *Klebsiella* sp. Sal 1, and *Enterobacter* sp. Sal 3, isolated from Nepalese sweet potato [15], were used in this study (Table 1).

**Table 1.** Bacterial isolates used in this study [5,6].

| Strain | Most Similar Genus [a] | Class | Origin | Accession Number |
|--------|------------------------|-------|--------|------------------|
| OYK | *Bacillus* sp. | Bacilli | Soil | LC590219 |
| RF-12 | *Bacillus* sp. | Bacilli | Rhizosphere | LC593252 |
| RF-37 | *Bacillus* sp. | Bacilli | Rhizosphere | LC593253 |
| F-33 | *Bacillus* sp. | Bacilli | Endophytic | LC430058 |
| Sal 1 | *Klebsiella* sp. | γ-Proteobacteria | Endophytic | LC389410 |
| Sal 3 | *Enterobacter* sp. | γ-Proteobacteria | Endophytic | LC389433 |
| Sal 6 | *Herbaspirillum* sp. | β-Proteobacteria | Endophytic | LC389442 |

[a] Based on the 16S rRNA gene sequence in the database.

### 2.2. Plant Growth Promotion and Colonization of Bacillus sp. Strains in Tomato Plant

To prepare the bacterial inoculum, each *Bacillus* sp. strain was cultivated in Modified Rennie (MR) [31] liquid medium with shaking at 150 rpm at 26 °C for 3 days. The culture was washed twice with sterilized distilled water by centrifugation at $10000 \times g$ at 4 °C for 10 min, and the cell pellet was resuspended with sterilized distilled water at $10^8$ colony

forming units (CFU)/mL to prepare an inoculum based on OD–CFU/mL correlated linear equations prepared for each strain.

In this study, we used tomato as a test plant due to the difficulty in preparing bacteria-free plants in sweet potato. Tomato seeds (*Solanum lycopersicum* L. cv. Chika F1 hybrid, Takii & Co., Ltd., Kyoto, Japan) were surface sterilized with 70% ethanol for 1 min followed by 1% sodium hypochlorite with 3–4 drops of Tween-20 for 13 min and washed 7–8 times with sterilized distilled water. The seeds were sown in the sterilized vermiculite in a Leonard jar [32] supplied with the sterilized Hoagland solution [33], and 1 mL of the inoculum was added onto the seed zone. The jar was put in a ventilated (<0.2 mm pore size) transparent plastic bag (Sun bag, Sigma-Aldrich, Tokyo, Japan), and after thinning out to one plant per jar, the tomato plant was aseptically cultivated in a phytotron (Model-LH 220S, Nippon Medical & Chemical Instruments Co., Ltd., Osaka, Japan) at 28/25 °C (16h/8h, day/night) for 24 days. An autoclaved culture was used as a control, and the experiment was conducted twice, using three plants for each treatment.

After cultivation, the tomato crop was harvested, and the fresh weight and length of the root and shoot were measured. Then, the population of the inoculated strains in the root, shoot, and rhizosphere was determined using two plants for each treatment. A rhizosphere sample was prepared by dipping and gently shaking the roots in sterilized distilled water. After washing the plant surface 6–7 times with sterilized distilled water, the root and shoot samples were separated and macerated with sterilized distilled water using a sterilized mortar and pestle, and the samples were subjected to dilution plating for the determination of CFU/g. At the same time, an aliquot of the final washing solution was directly plated, and no colony was observed. The inoculation experiment was conducted twice.

### 2.3. Effect of Co-Inoculation on Plant Growth Promotion and Colonization of Bacillus sp. F-33 with the Other Endophytic Strains in Tomato Plant

*Bacillus* sp. F-33 was used as a representative of the *Bacillus* sp. strains with the other endophytic strains, *Klebsiella* sp. Sal 1, *Enterobacter* sp. Sal 3, and *Herbaspirillum* sp. Sal 6, to examine the effect of co-inoculation on their plant growth promotion and colonization in the tomato plant.

Each bacterial strain was cultivated under the same conditions as described in Section 2.2 to prepare the inoculum at ca. $10^8$ CFU/mL. In case of co-inoculation, the same volume of individual cell suspension was mixed. The sterilized seeds were sown in the sterilized vermiculite in a capped glass tube (12 cm $\times$ 3 cm) supplied with the sterilized Hoagland solution, and 1 mL of the inoculum was added onto the seed zone. The other procedures were the same as those described in Section 2.2 except that the cultivation period was 14 days, and that the plant samples were macerated using a BioMasher (Nippi, Tokyo, Japan). The morphologies of the colonies of the co-inoculated strains were clearly different for counting separately. The inoculation experiment was conducted twice.

### 2.4. Effect of Time Interval Inoculation on Plant Growth Promotion and Colonization of Bacillus sp. F-33 and Klebsiella sp. Sal 1 in Tomato Plant

*Bacillus* sp. F-33 and *Klebsiella* sp. Sal 1 were used as representatives of the *Bacillus* sp. and the endophytic strains, respectively, to examine the effect of time interval of inoculation on their plant growth promotion and colonization in the tomato plant. The experimental procedures were the same as those described in Section 2.3 except that *Bacillus* sp. F-33 was inoculated first, and then *Klebsiella* sp. Sal 1 was separately inoculated 7 days after the first inoculation. The tomato plants were harvested at 14 days after the first inoculation. An experiment with a different order of inoculation, *Klebsiella* sp. Sal 1 first and *Bacillus* sp. F-33 s, was also conducted in the same way. The inoculation experiment was conducted twice, but one experiment was done using two plants and one of the plants was used to determine the population.

*2.5. Statistical Analysis*

Statistical analysis of the data on the plant growth and population of the inoculant obtained in each twice-repeated experiment was performed using the MSTAT-C 6.1.4 [34] software package. Data were subjected to Tukey's test after one-way ANOVA.

## 3. Results

### 3.1. Plant Growth Promotion and Colonization of Bacillus sp. Strains in Tomato Plant

The effects of inoculation of the *Bacillus* sp. strains on the growth of the tomato plant are presented in Figure 1. All *Bacillus* sp. strains except for *Bacillus* sp. RF-37 showed plant growth promotion. The root and shoot weights, and the shoot lengths of the inoculated tomato plant were significantly larger than the control while the root lengths were not affected. More lateral root development was observed in the inoculated tomato plant compared with the control.

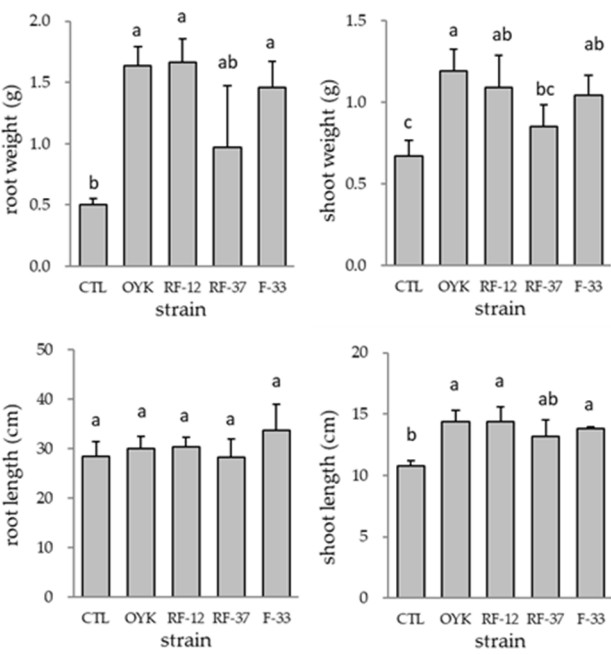

**Figure 1.** The effects of inoculation of *Bacillus* sp. strains on the growth of the tomato plant. The tomato plant was cultivated using sterilized vermiculite, and the parameters were measured at 24 days after seed inoculation. CTL represents the control samples inoculated with autoclaved cultures. The bars represent the standard deviation (n = 6), and different letters indicate significant differences at $p < 0.05$ by Tukey's test.

The populations of the inoculated *Bacillus* sp. strains in the rhizosphere, root, and shoot of the tomato plants are presented in Figure 2. All *Bacillus* sp. strains were detected in the rhizosphere, root, and shoot, and the populations of *Bacillus* sp. RF-12 and RF-37, which were originally isolated from the rhizosphere of sweet potato, and that of *Bacillus* sp. F-33, which was originally isolated as an endophyte of sweet potato, were higher than that of *Bacillus* sp. OYK, which was originally isolated from soil. The populations of the three *Bacillus* sp. strains were 0.9–2.2, 2.1–2.8, and 1.4–2.2 orders higher than those of *Bacillus* sp. OYK in the rhizosphere, root, and shoot, respectively. The populations were 2.4–4.0 and 3.1–5.2 orders higher in the rhizosphere than those in the root and shoot, respectively. No colony appeared in the control samples.

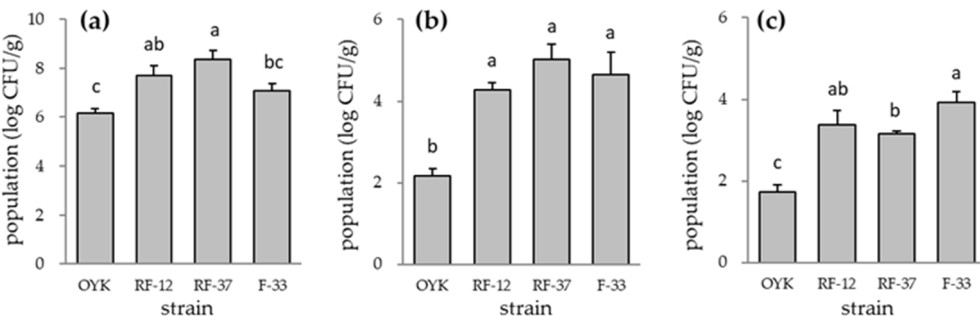

**Figure 2.** Colonization of seed-inoculated *Bacillus* sp. strains in the rhizosphere (**a**), root (**b**), and shoot (**c**) of the tomato plant. The tomato plant was cultivated using sterilized vermiculite, and colonization was examined at 24 days after seed inoculation. No colony appeared in the control samples. The bars represent the standard deviation (n = 4), and different letters indicate significant differences at $p < 0.05$ by Tukey's test.

### 3.2. Effect of Co-Inoculation on Plant Growth Promotion and Colonization of Bacillus sp. F-33 with the Other Endophytic Strains in Tomato Plant

The effects of co-inoculation of *Bacillus* sp. F-33 with the other endophytic strains are presented in Figure 3. The plant growth tended to be promoted by *Bacillus* sp. F-33 but not significantly. The reduction tendencies of the effects were observed by co-inoculation of *Enterobacter* sp. Sal 3 and *Herbaspirillum* sp. Sal 6. In shoot weight and root length, the effects of the co-inoculation seemed to be negative in most cases.

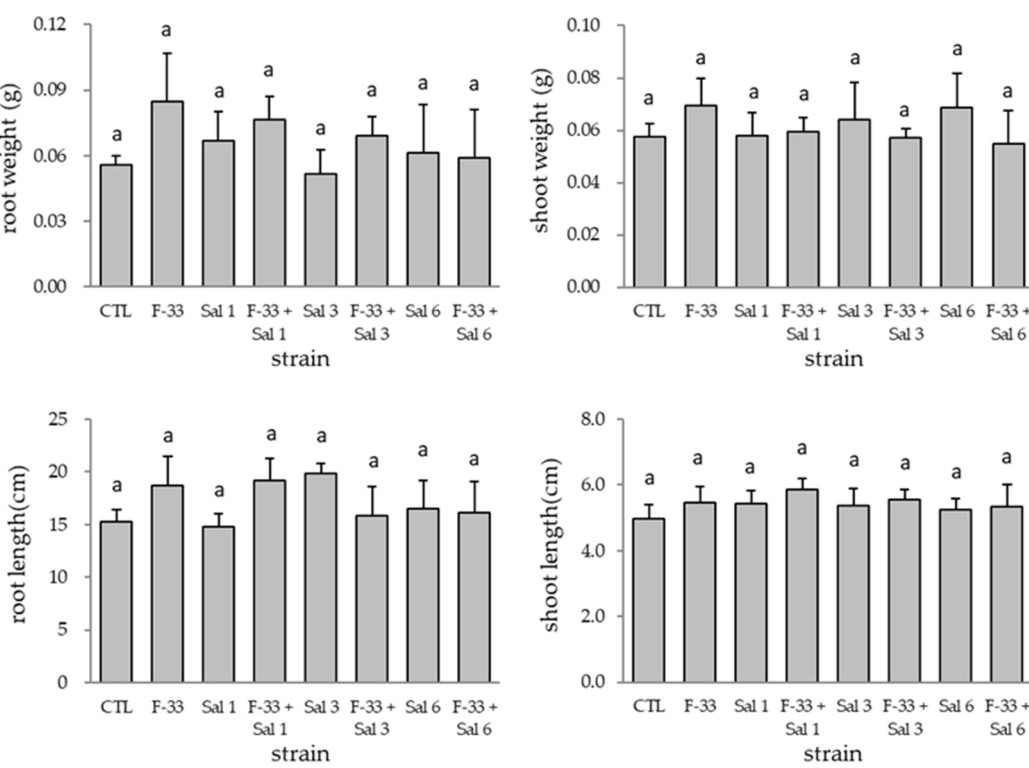

**Figure 3.** The effects of co-inoculation of *Bacillus* sp. F-33 with the other endophytic strains, *Klebsiella* sp. Sal 1, *Enterobacter* sp. Sal 3, and *Herbaspirillum* sp. Sal 6, on the growth of the tomato plant. The tomato plant was cultivated using sterilized vermiculite, and the parameters were measured at 14 days after seed inoculation. CTL represents the control samples inoculated with autoclaved cultures. The bars represent the standard deviation (n = 6), and different letters indicate significant differences at $p < 0.05$ by Tukey's test.

All strains colonized tomato plants, resulting in a large population, in which those of the endophytic strains were 1.5–1.7, 1.7–2.6, and 1.2–2.3 orders higher than those of *Bacillus* sp. F-33 in the rhizosphere, root, and shoot, respectively (Figure 4). Among the endophytic strains, the populations were not different in the rhizosphere, but the populations of *Herbaspirillum* sp. Sal 6 were about one order of magnitude higher than *Klebsiella* sp. Sal 1 in the plant parts. The populations were 1.8–2.7 and 2.3–3.3 orders higher at the rhizosphere than those in the root and shoot, respectively. No colony appeared in the control samples.

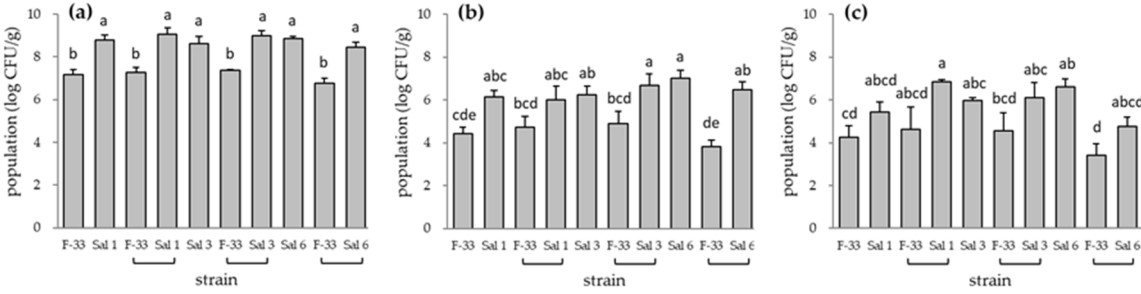

**Figure 4.** The effect of the seed-co-inoculated *Bacillus* sp. F-33 with the other endophytic strains, *Klebsiella* sp. Sal 1, *Enterobacter* sp. Sal 3, and *Herbaspirillum* sp. Sal 6, on colonization in the rhizosphere (**a**), root (**b**), and shoot (**c**) of the tomato plant. The tomato plant was cultivated using sterilized vermiculite, and colonization was examined at 14 days after seed-co-inoculation. The bracket on the x-axis indicates each population in co-inoculation, and no bracket indicates single inoculation. No colony appeared in the control samples. The bars represent the standard deviation (n = 4), and different letters indicate significant differences at $p < 0.05$ by Tukey's test.

In case of the co-inoculation, no apparent change in the population was observed in most cases. In co-inoculation of *Bacillus* sp. F-33 and *Herbaspirillum* sp. Sal 6, however, the population in the shoot tended to decrease by 0.8 and 1.8 orders in *Bacillus* sp. F-33 and *Herbaspirillum* sp. Sal 6, respectively. In addition, one example of a positive tendency in the co-inoculation was observed in the population of *Klebsiella* sp. Sal 1 in the shoot, in which a 1.4-order increase was observed.

*3.3. Effect of Time Interval Inoculation on Plant Growth Promotion and Colonization of Bacillus sp. F-33 and Klebsiella sp. Sal 1 in Tomato Plant*

The effects of the time interval of inoculation of *Bacillus* sp. F-33 and *Klebsiella* sp. Sal 1 are presented in Figure 5. The plant growth seemed to be promoted by *Bacillus* sp. F-33 but not by *Klebsiella* sp. Sal 1. When *Klebsiella* sp. Sal 1 was inoculated after *Bacillus* sp. F-33, the plant growth-promoting effects tended to be reduced in root weight. On the other hand, when *Bacillus* sp. F-33 was inoculated after *Klebsiella* sp. Sal 1, the effects seemed to be increased compared with the single inoculation of *Klebsiella* sp. Sal 1.

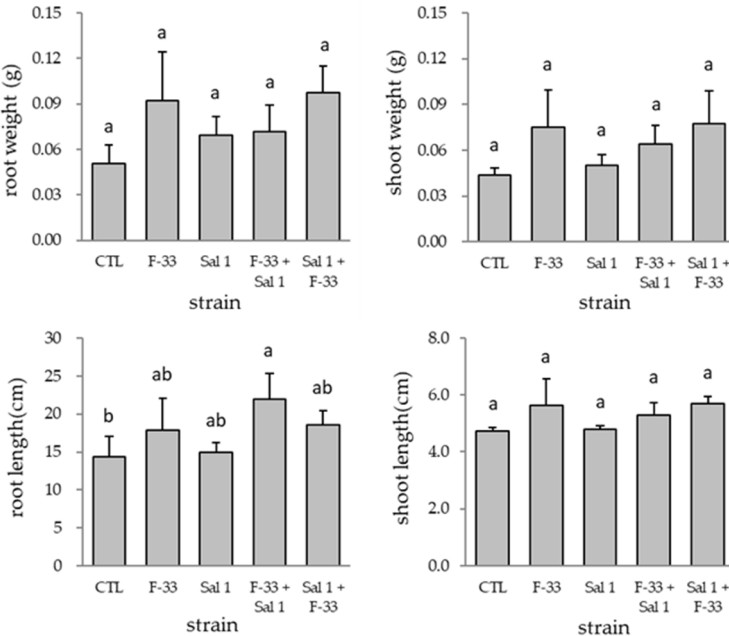

**Figure 5.** The effects of the time interval of inoculation on plant growth promotion and colonization of *Bacillus* sp. F-33 and *Klebsiella* sp. Sal 1 in the tomato plant. The tomato plant was cultivated using sterilized vermiculite, and the parameters were measured at 14 days after seed inoculation. In the time interval of inoculation, F-33 + Sal 1 and Sal 1 + F-33, the second inoculation was conducted 7 days after the first inoculation and analyzed 7 days after the second inoculation. CTL represents the control samples inoculated with autoclaved cultures. The bars represent the standard deviation (n = 5), and different letters indicate significant differences at $p < 0.05$ by Tukey's test.

In individual inoculation, populations of *Klebsiella* sp. Sal 1 were 1.9, 1.7, and 3.0 orders higher than those of *Bacillus* sp. F-33 in the rhizosphere, root, and shoot, respectively, and the populations were 2.7–2.8 and 2.5–3.7 orders higher in the rhizosphere than those in the root and shoot, respectively (Figure 6). When *Klebsiella* sp. Sal 1 was inoculated after *Bacillus* sp. F-33, the populations of *Bacillus* sp. F-33 were similar to those in the individual inoculation. When *Bacillus* sp. F-33 was inoculated after *Klebsiella* sp. Sal 1, those were 1.3–2.4 orders lower than those in individual inoculation. The populations of *Klebsiella* sp. Sal 1 showed similar levels under any conditions. No colony appeared in the control samples.

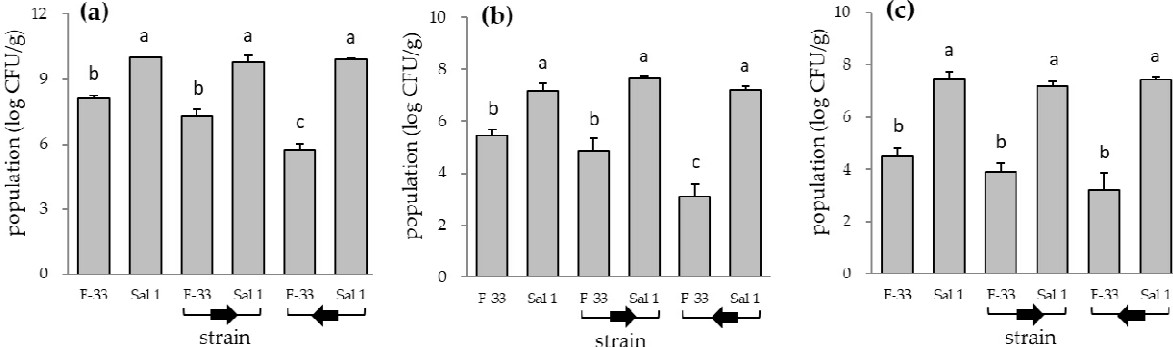

**Figure 6.** The effect of the time interval of inoculation of *Bacillus* sp. F-33 and *Klebsiella* sp. Sal 1 on colonization in the rhizosphere (**a**), root (**b**), and shoot (**c**) of the tomato plant. The tomato plant was cultivated using sterilized vermiculite, and colonization was examined at 14 days after seed inoculation. In the time interval of inoculation, F-33 + Sal 1 and Sal 1 + F-33, the second inoculation was conducted 7 days after the first inoculation and analyzed 7 days after the second inoculation. The bracket on the x-axis indicates each population in the time interval of inoculation, and the arrows on the bracket indicate the order of inoculation. No bracket indicates a single inoculation. No colony appeared in the control samples. The bars represent the standard deviation (n = 3), and different letters indicate significant differences at $p < 0.05$ by Tukey's test.

## 4. Discussion

Significant plant growth-promoting properties were observed in the *Bacillus* sp. strains except for *Bacillus* sp. RF-37 (Figure 1). Similar PGPR properties in *Bacillus* spp. have been previously reported [35–38]. In this study, the inoculants stimulated lateral root growth, resulting in greater root weight, which could explain the inconsistent results on root weight and root length in the inoculated plants. As indole-3-acetic acid (IAA) is known to have similar effects on plants [39], the plant growth promotion might be caused by IAA production by the inoculants. In another experiment, *Bacillus* sp. RF-12 and F-33 showed an IAA-producing ability while *Bacillus* sp. RF-37 did not (data not shown). However, since *Bacillus* sp. OYK also showed no activity, the reason for the plant growth promotion is unclear.

In addition to the IAA production, other tomato plant growth-promoting mechanisms by *Bacillus* spp. strains have been reported as follows: gibberellic acid (GA3) as well as IAA production [37,40,41], organic acid production and phosphate-solubilizing abilities [37,40,41], siderophores production [37,42], nitrogen fixation [37], and 1-aminocyclopropane-1-carboxylate (ACC) deaminase production [37,42].

In our previous study, the inoculated *Bacillus* sp. OYK strain could not establish its population as an endophyte in sweet potato [14], although *Bacillus* spp. strains have been reported as indigenous endophytes in sweet potato [15,16], tomato [19], banana [20], and switchgrass [21]. We attributed it to the competition with indigenous rhizobacteria and endophytes, as well as the endophytic ability of the inoculant.

In this study, all *Bacillus* strains colonized in the rhizosphere and endosphere of the tomato plants cultivated using sterilized vermiculite (Figure 2), suggesting that *Bacillus* sp. OYK has endophytic potential, and that the presence of indigenous microorganisms inhibited its colonization. However, the 1.4–2.8-orders lower populations of *Bacillus* sp. OYK in the plants compared with the other *Bacillus* sp. strains, which were isolated from the rhizosphere or as an endophyte (Figure 2), suggests decreased competitiveness of *Bacillus* sp. OYK against indigenous plant-associated microbes. Some genes and functions may be involved in the plant colonization ability, and PGPR strains from different habitats may have different interactions with plants. The use of originally plant associated PGPR could establish their populations at the rhizosphere and/or endosphere of plants.

The plant growth-promoting effects of *Bacillus* sp. F-33 were reduced in the presence of the other endophytes, though the population of *Bacillus* sp. F-33 was maintained

(*Klebsiella* sp. Sal 1 and *Enterobacter* sp. Sal 3) or slightly decreased (*Herbaspirillum* sp. Sal 6) (Figures 3 and 4), suggesting that its phyto-stimulating ability was neutralized by the other strains. As the three co-inoculated strains have IAA-degrading ability [30], they might degrade IAA produced by *Bacillus* sp. F-33 below the effective level.

Synergetic effects of co-inoculation have been reported [23–26] while cancelation of the positive effects [43–45], and negative effects of co-inoculation have also been reported [29,30]. The effects of the co-inoculation seemed to be dependent on the combination of the strains. In most studies that examined the effects of co-inoculation of PGPR, changes in populations of the PGPR by co-inoculation were not measured. In the limited examples of the study using *Azospirillum brasilense* Sp245 and *Bacillus subtilis* 101 [29], and *Klebsiella* sp. Sal 1 and *Herbaspirillum* sp. Sal 6 [30], their plant growth promotions were reduced even though the populations of the PGPR were maintained, as observed in this study. In our previous study, diverse endophytic bacterial communities were observed in sweet potato, and some components of the communities disappeared by inoculation of *Bacillus* sp. OYK [14]. It is crucial to elucidate the mechanisms of the microbial interactions; however, it might be complex given the actual environment.

After the establishment of *Bacillus* sp. F-33 in the rhizosphere and in the tomato plant, *Klebsiella* sp. Sal 1 could colonize the same population as the strain was individually inoculated (Figure 6) and inhibited the plant growth-promoting ability of *Bacillus* sp. F-33 without reducing its population (Figure 5), as in the co-inoculation experiment. The high colonizing potential of *Klebsiella* sp. Sal 1 seemed not to be affected by the about 2-orders lower population of the previously established *Bacillus* sp. F-33.

On the other hand, after the establishment of *Klebsiella* sp. Sal 1, the colonization of *Bacillus* sp. F-33 was reduced by 1.3–2.4 orders than those in the individual inoculation (Figure 6). The relatively lower potential for colonization of *Bacillus* sp. F-33 might be the reason. The microbial community structure might be a crucial factor to determine the fate of allochthonous microorganisms, such as a PGPR inoculant. Pre-inoculation of PGPR prior to transplantation could be one practical method to enhance higher colonization in plants.

In spite of the reduced population of *Bacillus* sp. F-33, the plant growth promotion was increased when the strain was inoculated after *Klebsiella* sp. Sal 1 (Figure 5). It was suggested that the level of the population is not a determinant of the potential of the strain. Although the population of *Bacillus* sp. F-33 was maintained both in the co-inoculation and in the inoculation of *Klebsiella* sp. Sal 1 after *Bacillus* sp. F-33, the PGPR potential of *Bacillus* sp. F-33 was reduced in the presence of *Klebsiella* sp. Sal 1, so unknown factors might be involved in plant growth promotion. In addition, the ratio between the populations might not be constant when plants developed, and the kinetic of the different bacterial populations might not be reflected by one sampling time. Time course analysis after inoculation could reveal the progress of colonization in the plant. The results of this study also indicate that there are different niches for the different strains and the colonization of these niches may not have the same impact on plant growth. It may mean that bacteria are competing for some niche colonization.

In addition to plant growth-promoting properties, the colonization potential should be considered as important criteria when assessing their suitability for commercial development. The lower population of *Bacillus* sp. OYK, which was isolated from soil, than the other *Bacillus* sp. strains, which were isolated from either the rhizosphere or endosphere of plant samples, suggests the importance of the origin of the strains for their colonization. The plant growth promotion and colonization potentials were independently affected by the co-existing microorganisms. Further studies are necessary to evaluate the colonization potential of PGPR under field conditions where diverse microorganisms exist.

## 5. Conclusions

In this study, the higher population of rhizospheric and endophytic *Bacillus* sp. in the plant suggest the importance of the origin of the strains for their colonization. The plant

growth promotion and colonization potentials were independently affected by the co-existing microorganisms.

**Author Contributions:** A.S. and K.I. conceptualized the study and designed the experiments; A.S. performed the experiments; R.R.P. isolated rhizospheric and endophytic strains; M.H.R.H. assisted data analysis; A.S. wrote the article, with a substantial contribution from K.I. All authors have read and agreed to the published version of the manuscript.

**Funding:** This research received no external funding.

**Conflicts of Interest:** The authors declare no conflict of interest.

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
