# Peer review of "Effect of Co-Inoculation of Bacillus sp. Strain with Bacterial Endophytes on Plant Growth and Colonization in Tomato Plant (Solanum lycopersicum)"

_2036-7481, doi:10.3390/microbiolres12020032_

Round 1
Reviewer 1 Report
The manuscript entitled “Effect of Co-inoculation of Bacillus sp. Strain with Bacterial Endophytes on Plant Growth and Colonization in Tomato Plant (Solanum lycopersicum)” presents the evaluation of the colonization potential of the Bacillus sp. strains with different origin in tomato.
The manuscript is very interesting and well-written. However, minor revisions should be made in order to be published in Microbiology Research journal, and the manuscript should be completed and/or modified taking into account the suggestions below:
- The authors are advised to rephrase the sentences from lines: 19-20, 38-40, 137-139, 250-253, 262-264, 306-309.
- The authors are advised to use the same abbreviation for liter, for consistency (see lines 91, 98, etc)
- The authors are advised to present the references for section 2. Materials and Methods (subsections 2.2-2.4)
- The authors are advised to present the results in tables, too.
- The authors are advised to use Italic style for Bacillus (line 260)
Author Response
Thank you for your valuable comments for improving our manuscript. According to the reviewer’s suggestions, we have revised our manuscript as follows:
The authors are advised to rephrase the sentences from lines: 19-20, 38-40, 137-139, 250-253, 262-264, 306-309.
We have rephrased the sentences as follows:
After cultivation in a phytotron, plant growth parameters and populations of the inoculants in the root, shoot, and rhizosphere were determined. (19-20)
Plant growth-promoting rhizobateria (PGPR) are becoming more widely accepted in intensive agriculture to enhance sustainable agricultural production in various parts of the world [1]. (38-40)
The inoculation experiment was conducted twice, but one experiment was done using two plants and one of the plants was used to determine the population. (139-141)
In another experiment, Bacillus sp. RF-12 and F-33 showed IAA-producing ability while Bacillus sp. RF-37 did not (data not shown). However, since Bacillus sp. OYK also showed no activity, the reason for the plant growth promotion is unclear. (252-255)
We attributed it to the competition with indigenous rhizobacteria and endophytes, as well as the endophytic ability of the inoculant. (264-265)
Although the population of Bacillus sp. F-33 was maintained both in the co-inoculation and in the inoculation of Klebsiella sp. Sal 1 after Bacillus sp. F-33, the PGPR potential of Bacillus sp. F-33 was reduced in the presence of Klebsiella sp. Sal 1, so unknown factors might be involved in the plant growth promotion. (310-313)
The authors are advised to use the same abbreviation for liter, for consistency (see lines 91, 98, etc)
We corrected. (92)
The authors are advised to present the references for section 2. Materials and Methods (subsections 2.2-2.4)
The procedures for this study were applied without reference.
The authors are advised to present the results in tables, too.
We think that the table presentation would be redundant and not necessary.
The authors are advised to use Italic style for Bacillus (line 260)
We corrected. (262)
Reviewer 2 Report
I appreciate the hard work in preparing the experiments and presenting in this paper the results you have obtained. The paper is clear and concise, therefore, I have only few suggestions as follows:
- I didn't find the aim of the experiments. What did you wanted to obtain by adding PGPR: higher crops, height quality tomatoes, etc.?
- Why have you chosen Bacillus sp. F-33 and not the others, or Klebsiella sp. Sal-1 in the experiments described in 2.3 and 2.4 (Material and method). This information is missing from the section.
- Even more conclusion can be read between the lines in the Discussion section, you can add them in the appropriate part of the paper. Eg.: Which one of the bacterial strains tested individually or combined got established faster and in which area? Which PGPR of those mentioned in the paper has the bigger influence on increasing the yield, or quality of fruits, etc...?
- the conclusion you have written was revealed first on your previous study on sweet potato.

Author Response
Thank you for your valuable comments for improving our manuscript. According to the reviewer’s suggestions, we have revised our manuscript as follows:
I didn't find the aim of the experiments. What did you wanted to obtain by adding PGPR: higher crops, height quality tomatoes, etc.?
We revised introduction to clarify the aim of this study. (67-75)
Why have you chosen Bacillus sp. F-33 and not the others, or Klebsiella sp. Sal-1 in the experiments described in 2.3 and 2.4 (Material and method). This information is missing from the section.
We used them as representatives of the Bacillus and endophytic strains, respectively, and there is no particular reasons for the selection. (117, 132-133)
Even more conclusion can be read between the lines in the Discussion section, you can add them in the appropriate part of the paper. Eg.: Which one of the bacterial strains tested individually or combined got established faster and in which area? Which PGPR of those mentioned in the paper has the bigger influence on increasing the yield, or quality of fruits, etc...?
Because we examined only limited numbers of combination of PGPR and endophytes in the laboratory experiment using the sterilized vermiculite, and because the samples were taken at one point during the cultivation, it seems to be hard to mention about the reviewer’s comments. However, in response to the reviewer’s suggestion, we added a few sentences in Discussion. (274-275, 304-306, 315-316)
the conclusion you have written was revealed first on your previous study on sweet potato.
We arranged them in the order of the aim of the study presented in Introduction.
This manuscript is a resubmission of an earlier submission. The following is a list of the peer review reports and author responses from that submission.